# A Preliminary Study of Mild Heat Stress on Inflammasome Activation in Murine Macrophages

**DOI:** 10.3390/cells12081189

**Published:** 2023-04-19

**Authors:** Simmie L. Foster, Abigail J. Dutton, Adina Yerzhan, Lindsay B. March, Katherine Barry, Corey R. Seehus, Xudong Huang, Sebastien Talbot, Clifford J. Woolf

**Affiliations:** 1Department of Psychiatry, Massachusetts General Hospital and Harvard Medical School, Boston, MA 02114, USA; 2FM Kirby Neurobiology Center, Boston Children’s Hospital and Department of Neurobiology, Harvard Medical School, Boston, MA 02115, USA; 3Department of Pharmacology and Physiology, Karolinska Institutet, SE-171 77 Stockholm, Sweden; 4Department of Biomedical and Molecular Sciences, Queen’s University, Kingston, ON K7L 3N6, Canada

**Keywords:** mild heat stress, caspase 1, inflammasome, cytokine, IL-1β, mitochondria, pyrexia, hyperthermia

## Abstract

Inflammation and mitochondrial-dependent oxidative stress are interrelated processes implicated in multiple neuroinflammatory disorders, including Alzheimer’s disease (AD) and depression. Exposure to elevated temperature (hyperthermia) is proposed as a non-pharmacological, anti-inflammatory treatment for these disorders; however, the underlying mechanisms are not fully understood. Here we asked if the inflammasome, a protein complex essential for orchestrating the inflammatory response and linked to mitochondrial stress, might be modulated by elevated temperatures. To test this, in preliminary studies, immortalized bone-marrow-derived murine macrophages (iBMM) were primed with inflammatory stimuli, exposed to a range of temperatures (37–41.5 °C), and examined for markers of inflammasome and mitochondrial activity. We found that exposure to mild heat stress (39 °C for 15 min) rapidly inhibited iBMM inflammasome activity. Furthermore, heat exposure led to decreased ASC speck formation and increased numbers of polarized mitochondria. These results suggest that mild hyperthermia inhibits inflammasome activity in the iBMM, limiting potentially harmful inflammation and mitigating mitochondrial stress. Our findings suggest an additional potential mechanism by which hyperthermia may exert its beneficial effects on inflammatory diseases.

## 1. Introduction

Multiple disorders that affect cognition and mood appear to have an inflammatory basis. For example, Alzheimer’s disease (AD), major depressive disorder (MDD), and even the neurological symptoms of the post-acute sequelae of SARS-CoV-2 (PASC) have been associated with dysregulated inflammation and increased oxidative stress [1,2,3]. This is not surprising, as inflammation and oxidative stress are also related, with mitochondria-dependent reactive oxygen species (ROS) playing a key role in host defense as well as contributing to the dysregulated inflammation of aging (inflammaging) [4]. Consistently, anti-inflammatory/antioxidant treatments have shown some promise for treating neuroinflammatory disorders, and there is an interest in the development of new treatments [4].

Mild whole-body hyperthermia (WBH), or elevating the body temperature to that of a low-grade fever (38–39 °C), is a promising anti-inflammatory treatment that has been shown to improve mood and may even prevent dementia [5,6]. Indeed, inflammatory ailments such as depression [5], AD [6], and various malignancies [7] have been reported to respond to hyperthermic treatment. Heat therapy has even been proposed to treat the SARS-CoV-2 (COVID-19) infection [8]. The molecular mechanisms underlying the treatment effect of heat exposure are not well understood but are likely mediated by monocytic cells/macrophages (the predominant producers of proinflammatory cytokines in the body). Previous studies modeling hyperthermia in a dish indicated that exposing macrophages in vitro to a heat shock of 40–43 °C can inhibit inflammatory cytokines [9,10,11,12]. However, few studies have addressed the effect of a mild heat shock of 38–39 °C, which would be more relevant to the current clinical paradigm. In addition, little is known about the effect of mild heat shock on downstream inflammatory mediators, such as the inflammasome, a molecular complex that orchestrates inflammation, modulates mitochondrial function, and induces oxidative stress and has a key role in the pathogenesis of AD, MDD, and PASC [3,13,14].

Inflammasomes typically contain three main components: a sensor protein (such as the NOD-, LRR-, and pyrin domain-containing protein 3, or NLRP3), an adaptor protein, apoptosis-associated speck-like protein containing a caspase activation and recruitment domain, or ASC, and an inflammatory caspase, typically caspase 1 [15]. Inflammasomes are substantially expressed in macrophages [16,17], are classified according to the sensor protein they contain, and require two signals for full activation. For example, for the NLRP3-containing inflammasome, activation requires a priming signal such as LPS, which induces the transcription of inflammasome components (NLRP3) and pro-IL-1β [15]. This pro-IL-1β accumulates intracellularly and is not functional. A secondary signal, such as ATP or nigericin, induces the assembly of the inflammasome and leads to the autocatalysis of procaspase 1. In turn, the newly formed caspase 1 cleaves pro-IL-1β into IL-1β as well as acting on other proteins, including gasdermin D (GSDMD), required for IL-1β release [18].

There is some indication the inflammasome may be sensitive to temperature regulation, for example, a hyperactive mutation in NLRP3 is responsible for familial cold autoinflammatory syndrome, in which aberrant activation of the inflammasome is triggered by cold [19]. However, the regulation of the inflammasome and impact on mitochondrial function by heat is still not well understood. Here we present a preliminary study on the effect of mild heat shock on LPS-stimulated inflammatory cytokine production, inflammasome activation, and mitochondrial activity in immortalized bone marrow-derived murine macrophages (iBMM). 

## 2. Materials and Methods

### 2.1. Cell Lines, Cell Culture, and Reagents

Immortalized bone marrow-derived macrophages (iBMMs) and iBMMs expressing ASC-citrine were developed in the laboratory of Jonathan Kagan through retroviral transduction of primary murine bone marrow-derived macrophages from WT mice or transgenic mice expressing ubiquitous CRE on a WT ASC background (transgenic mice developed in the Golenbock lab) [18,20] and were a kind gift from Jonathan Kagan and Charles Evavold. Cells were cultured in Dulbecco’s Modified Eagle Medium (DMEM) containing 10% Heat-Inactivated Fetal Bovine Serum (FBS), penicillin, streptomycin, L-glutamine, sodium pyruvate, and β-mercaptoethanol. Cells were washed in phosphate buffered saline (PBS) pH 7.4 containing 2 mM EDTA to detach cells for passage. Cells were passaged 1:10 every 3 days. Lipopolysaccharide (LPS) from *E. coli* serotype O55:B5 (L2280, Sigma Aldrich, St. Louis, MO, USA) was reconstituted at 1 mg/mL, aliquoted, and frozen at −80 °C for later use. The 5′adenosine triphosphate (ATP, A6419, Sigma Aldrich) was resuspended at 500 mM, adjusted to pH 7.4, and stored at −20 °C. Propidium iodide (PI, P4864, Sigma Aldrich) was purchased in a ready-to-use stock solution at a concentration of 1 mg/mL.

### 2.2. LDH Assay, ELISAs

Cell culture supernatants were cleared of cells by spinning 24- or 96-well cell culture plates at 400× *g* for 5 min. Supernatants were transferred to a 96-well clear bottom assay plate and assayed for LDH release using the Pierce LDH Assay Kit (Thermo Fisher Scientific, Waltham, MA, USA; Cat. no. 88954) as per the manufacturer’s instructions; individual experiments were normalized to the “no stim” condition, and then results were pooled. Supernatants were also assayed using the Biolegend mouse ELISA kits (IL-1β cat. no. 432604, IL-6 cat. no. 431304, and TNFα cat. no. 430904, Biolegend, San Diego, CA, USA) per the manufacturer’s instructions. For assessment of intracellular (cell-associated) IL-1β, cells were lysed using 0.5% Triton X-100 in PBS, and the lysate was assayed by ELISA as described above.

### 2.3. Western Blot and Antibodies

After stimulations, cells were lysed in RIPA buffer+reducing agent, run on 4–12% NuPage gels under denaturing conditions, and transferred to membranes using the NuPage transfer system. Membranes were blotted for GAPDH (Abcam, Waltham, MA, USA, ab181602), HSP72 (Enzo Life Sciences, Farmingdale, NY, USA, ADI-SPA-810-D), IL-1β (R&D Systems, Minneapolis, MN, USA, AF-401-SP), and Caspase 1 (Adipogen, San Diego, CA, USA, AG-20B-0042-C100), and visualized using the LiCOR Odyssey imaging system.

### 2.4. Real-Time Cell Permeability Assay (PI Assay), Membrane Fluidity Assay

Cells were plated on optically clear, black-walled 96-well plates and stimulated as described in the text. Prior to the addition of ATP, media was aspirated, a solution containing 2× ATP (10 μM) and 2× PI (10 μM) was added, and cells were incubated at 37 °C for 10 min. Real-time incorporation of PI was monitored using a Perkin Elmer Ensight plate reader. The program settings were bottom reading fluorescence with an excitation wavelength of 530 nm and an emission wavelength of 617 nm.

### 2.5. Live Cell Imaging

iBMMs expressing ASC-citrine were plated on 96-well glass-bottom imaging plates, primed with 1 μg/mL of LPS for 3 h, then exposed to 39 °C for 1 h or left at 37 °C, and ATP (500 μM) was added for 30 min. Cells were then stained with MitoTracker Red CMXRos (Thermofisher, M7512) at a dilution of 1:5000 for 20 min and Hoescht stain (Thermofisher, H3570) at 1:10,000 for 10 min. Cells were washed in warm DMEM to remove excess stain. The cells were kept at 37 °C prior to imaging. Imaging was obtained using the ImageXpressMicro Confocal live cell imaging confocal system. ASC-citrine was obtained in the green channel, MitoTracker Red CMXRos in the red channel, and Hoescht stain in the blue channel. An algorithm was empirically designed by Dr. Lee Barrett using the custom module editor in MetaXpress 6 software to create masks defining threshold size and intensity for Hoescht-stained nuclei and MitoTracker-Red-stained mitochondria. In addition, numbers of ASC specks/DAPI nuclei were counted manually per 40× field, with greater than 5 fields analyzed per condition. 

### 2.6. Statistical Analysis

Data with error bars are represented as the mean ± SEM. Statistical significance was determined with a two-way ANOVA with Tukey’s multiple comparison test correction. Only probability (*p*) values less than 0.05 were considered statistically significant. Details of individual experiments are defined in the figure legends.

## 3. Results

### 3.1. Mild and Extreme Heat Shocks Inhibit Proinflammatory Cytokines

To begin our investigation into the anti-inflammatory mechanisms of mild heat shock, we set up a system of heat shock in immortalized bone marrow-derived murine macrophages (iBMM), which express inflammasome components and inflammatory cytokines with a pattern similar to that of primary bone marrow-derived macrophages [18]. We used LPS as a priming signal to induce transcription of cytokines and inflammasome components and ATP as a secondary signal to induce processing and secretion of IL-1β, and we exposed the cells to two different temperatures (41.5 °C, extreme; and 39 °C, mild) with two different regimens of heat exposure, either concurrent with the priming stimulus or prior to the secondary stimulus, after the cells had been sufficiently primed (Figure 1A). We first asked if exposure to an extreme heat shock of 41.5 °C at the same time as LPS stimulation would inhibit iBMM cytokine production as has been seen in previous studies [9,10,12,21] (Figure 1A, “no priming”). Exposure to this elevated temperature diminished LPS-stimulated production of the prototypical cytokines tumor necrosis factor α (TNFα), IL-6, and intracellular and secreted IL-1β (Figure 1B–D, hatched bars) as compared to production at 37 °C. Next, we asked if the same held true for a mild heat shock of 39 °C. Indeed, mild heat shock was also able to inhibit these cytokines (Figure 1E–G, hatched bars) as compared to controls incubated at 37 °C (black bars). 

### 3.2. Heat Shock after LPS Priming Differentially Inhibits Cytokines without Impacting Cell Death

Since heat shock concurrent with LPS stimulation inhibits transcription of proinflammatory cytokines, limiting our ability to ask questions about the effect of heat on downstream mediators, we next primed the cells with LPS for 3 h prior to heat exposure, allowing for the accumulation of intracellular cytokines and secretion of non-inflammasome dependent cytokines IL-6 and TNFα into the culture media. IL-6 and TNFα are readily translated into a mature form ready for secretion, while IL-1β is made as a pro-molecule that requires caspase 1 processing prior to secretion. Even after priming, we still observed substantial inhibition of both secreted and intracellular IL-1β, indicating additional post-transcriptional regulation of IL-1β by heat exposure (Figure 1B,E; checked bars). Priming resulted in decreased inhibition of IL-6 and TNFα after both mild and extreme heat shock, consistent with these cytokines being subject to a primarily transcriptional repression mechanism (Figure 1C,D; checked bars). 

As hyperthermia can kill tumor cells [7] and stimulation of the inflammasome by LPS+ATP can also lead to cell death, termed pyroptosis, we questioned if the decrease in cytokine production could be attributed to cell death. As such, we set out to test if heat shock treatment affects cultured iBMM cell viability by monitoring the release of lactate dehydrogenase (LDH), a protein released by dying necrotic cells [18,22]. We did observe some cell death in the LPS+ATP condition at 37 °C, consistent with inflammasome activation (Figure 1H,I; black bars). In the co-exposure group, we found significant iBMM cell death at both temperatures in the LPS+ATP condition (Figure 1H,I; hatched bars). In contrast, LPS-primed heat-exposed iBMMs remined viable after addition of ATP (Figure 1H,I; checked bars). Overall, a mild heat shock (39 °C) in LPS-primed iBMMs does not impact macrophage survival and has a limited effect on IL-6 and TNFα, while it abolishes the production of IL-1β. Therefore, we chose this regimen for our subsequent experiments.

### 3.3. Heat Shock Decreases Membrane Permeability

Inflammasome activation in macrophages is associated with two states: (1) pyroptosis, as mentioned above, a form of inflammatory cell death that underlies much of the tissue damage attributed to inflammasome activation [23] and is associated with a ruptured cytosolic membrane allowing LDH release, and (2) hyperactivation, signified by live cells with increased membrane permeability. This change in membrane permeability is thought to be due to the activation of gasdermin D (GSDMD) by the inflammasome, which then oligomerizes and forms pores in the membrane, allowing IL-1β but not LDH release into the extracellular space [18]. Pore-formation may be monitored by incorporation of the membrane-impermeable dye propidium iodide (PI) into the nucleus. We did find that PI was incorporated in iBMM treated with LPS+ATP at 37 °C, indicating pore formation (Figure 2C; black line). However, heat-exposed iBMM one hour prior to ATP (37 °C→39 °C) showed no increase in membrane permeability over controls (Figure 2C; red line), indicating they are protected from pore formation. Since changes in membrane fluidity could affect the assembly of signaling platforms and partially explain the decrease in pore formation in heat-shocked cells [24], we next examined iBMM membrane fluidity by measuring the fluorescence of a lipophilic probe that forms excimers as membrane fluidity increases. We found that, in comparison to cells cultured at 37 °C, 39 °C-exposed iBMM showed increased membrane fluidity (Appendix A).

### 3.4. Heat Shock Inhibits Caspase 1 Processing

Our observation that IL-1β is selectively inhibited after a mild heat shock, even in primed cells, and our further observation that heat shock protects from membrane permeabilization led us to consider the possibility that a mild heat shock might inhibit inflammasome activity, as IL-1β secretion and pore formation are dependent on processing by the inflammasome [18]. Inflammasome activation requires the assembly of several components, including caspase 1, NLRP3, and the scaffolding protein (ASC). As such, we set out to examine the levels of caspase 1 processing in heat-exposed iBMM and discovered that elevated temperature (39 °C) decreased caspase 1 and IL-1β processing in these cells (Figure 3A and Appendix A). Furthermore, the inhibition was rapid, with a significant decrease in processing happening after 15 min of heat exposure (Figure 3B and Appendix A). In one experiment, this decrease was transient, recovering within 60 min, indicating some variability in the timing of inhibition (Appendix A). In addition, heat shock reduced levels of NLRP3 transcript in LPS+ATP or LPS+nigericin-treated cells (Appendix A). Subsequently, we sought to test whether HSP72, a protein induced upon a temperature increase to 42–43 °C and reported to inhibit caspase 1 processing [25], would explain some of the effect triggered by elevated temperature (39 °C). However, we found that in our system, HSP72 was constitutively produced in iBMMs, even in the absence of heat shock (Appendix A), ruling out its potential role upstream of caspase 1 processing in our iBMMs.

### 3.5. Mild Heat Shock Inhibits ASC Speck Formation but Not Mitochondrial Function

We next asked if heat shock may be interfering with the initial assembly and activation of the inflammasome. Assembly of the inflammasome can be monitored by assessing the formation of oligomerized ASC specks [15]. We performed live-cell imaging of LPS-primed cells exposed to mild heat shock and found a significant decrease in ASC speck formation in heat-exposed iBMM cells as compared to controls (Figure 4A,B). Mitochondria play a key role in host defense through oxidative metabolism and may also directly impact inflammasome activation [16,26,27]. However, little is known about the effect of heat shock on mitochondrial function. Therefore, we asked if heat shock affects the basic functioning of iBMM mitochondria. Using a mitochondrial membrane potential-sensitive dye, we found that heat shock did not impair this measure of mitochondrial function. In fact, there were increased numbers of polarized mitochondria in heat exposed cells. (Figure 4C). Taken together, these results indicate that mild heat shock at 39 °C quickly inhibits activation of the NLRP3 inflammasome in iBMM, resulting in decreased IL-1β production with no impairment in mitochondrial function.

## 4. Discussion

In this preliminary study, we have found that exposure of iBMMs to a mild heat shock of 39 °C rapidly inhibited caspase 1 processing, ASC speck formation, and membrane permeability, thereby preventing secretion of IL-1β without impairing mitochondrial polarization (Figure 5). The specific inhibition of IL-1β may represent a mechanism for regulation of fever, as IL-1β is a major cytokine responsible for resetting the body’s thermostat in fever. If this is the case, once a sufficient level of fever is reached, the heat itself would inhibit the production of IL-1β, limiting the drive for pyrexia and limiting the potential for overwhelming inflammation.

Our findings fit in with previous observations that inflammasome function is sensitive to temperature, both hypothermic and hyperthermic [19,25,28,29,30,31,32,33]. Notably, mutations in NLRP3 (first discovered as cryopyrin) are responsible for a group of disorders known as cryopyrin-associated autoinflammatory syndromes (CAPS) [19]. Karasawa et al. recently discovered that mutant NLRP3 aggregates in the cold to nucleate inflammasome assembly and activation, explaining some of the characteristic cold-induced inflammation in a subtype of CAPS [19]. Consistently, severe hypothermia (4 °C) was associated with increased inflammasome activation in a model of cold-stress-induced liver injury in mice [30]. However, as opposed to severe hypothermia (4 °C), several studies have shown that mild hypothermia (32–34 °C) inhibits inflammasome function in WT cells and rats [29,31,33].

While many previous studies on the regulation of inflammation by elevated temperature have focused on the transcriptional regulation of NFκB-dependent cytokine genes by Heat Shock Factor 1 (HSF1) [9,10,12,21], only three studies have investigated the effect of elevated temperature on caspase 1 activity [25,28,32]. Ahn et al. examined inflammasome activation under hyperthermic as well as hypothermic conditions after stimulation with monosodium urate crystals (MSU) or canonical stimuli such as nigericin and found that 37 °C is the optimal temperature for inflammasome activation with canonical stimuli [28]. Levin et al. discovered that heat shock induced the association of pro-caspase 1 with a large complex that could inhibit its activity, and furthermore, that the inhibitory factor was present in cell lysate and titratable and was not dependent on new transcription or translation, which would indicate it is not an induced HSP [32]. Finally, Martine et al. found that although HSP72 directly inhibits caspase 1 and inflammasome assembly in non-heat-shocked cells, heat shock in HSP70-deficient cells was still able to inhibit caspase 1 [25]. Our study is consistent with these findings, implying that inhibitory activity is not completely dependent on HSP induction, as we did not see an induction of HSP72, rather it was constitutively expressed (Appendix A).

Given the above observations and in light of our current study, there are several mechanisms that could be at play. The inflammasome has previously been shown to be regulated in normothermic conditions by (1) mitochondrial ROS; (2) changes in ion flux; and (3) post-translational modifications [34]. As we saw, mild heat shock protected against mitochondrial dysfunction (Figure 4), so we might expect that heat shock decreases mitochondrial ROS. In a model of therapeutic hypothermia for ischemic myocardial injury, it was found that hypothermia induced activation of SIRT3, a histone deacetylase of the sirtuin family, which is known to participate in autophagy, preserve mitochondrial function, and decrease ROS [33]. Chemical inhibition of SIRT3 partially rescued inflammasome activation in myocardial cells exposed to hypothermia [33], implying that hypothermia induces SIRT3 activity, which then leads to inflammasome inhibition, possibly by decreasing ROS. Interestingly, a recent study found that, rather than simply decreasing ROS, short exposures to heat shock in HeLa cells produced low levels of ROS that seemed to activate autophagy and protect from the cytotoxicity associated with higher ROS production [35]. Whether SIRT3/autophagy may also play a role in our system of hyperthermia in macrophages remains to be explored in future studies.

In terms of ion flux, we observed a decrease in membrane permeability (Figure 2), which would limit the K^+^ efflux and Na^+^/Ca^2+^ influx needed for inflammasome activation [34]. We hypothesize there could be a direct impact of heat either on the membrane itself (for example, changing the fluidity, as we saw in Appendix A) or on the composition of lipid raft signaling complexes. Heat shock could also affect the activation state of ion channels that are sensitive to temperature, for example TRP channels. Another possibility is that this decreased permeability is downstream of caspase 1 inhibition, due to a decrease in GSDMD cleavage. These possibilities remain to be formally tested.

It has previously been found that caspase 1 undergoes extensive post-translational modifications, including phosphorylation, ubiquitination, and sumoylation [34]. Future experiments will map which of these modifications are temperature-sensitive and could potentially mediate temperature-dependent inhibition of the inflammasome. Additionally, it will be essential to explore whether similar regulation happens in primary cells and determine the impact of heat exposure on GSDMD and other inflammasome regulated proteins such as IL-18.

Many important chronic diseases, including depression [13], various malignancies [36], autoimmunity [37], and AD [3], have been linked to pathological activation of the inflammasome and increased production of IL-1β. In addition, severe acute COVID-19 as well as post-acute sequelae of SARS-CoV-2 (PASC), also called “long-haul COVID-19,” have been proposed to be a consequence of excessive inflammasome activation [14,38]. Our results provide a new avenue towards understanding temperature control of the inflammasome and may help in the effort to develop more effective hyperthermia treatments for patients with inflammasome-related pathologies.

## 5. Conclusions

In conclusion, our investigation has shown that mild and clinically relevant heat shock (which is more tolerable to patients than the severe heat shock often used in in vitro studies) specifically inhibits the inflammasome without causing mitochondrial dysfunction. Our study has implications for the use of hyperthermia to treat inflammasome-related pathologies such as depression, various malignancies, COVID-19, and AD.

## Figures and Tables

**Figure 1 cells-12-01189-f001:**
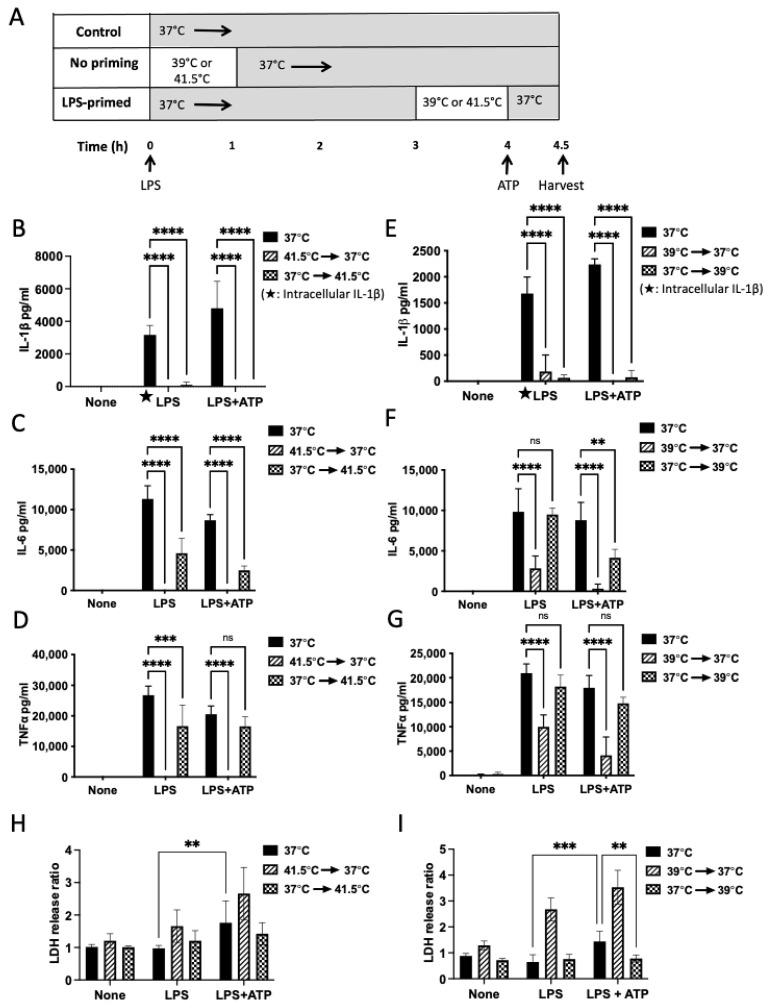
Mild and extreme heat shock differentially inhibit proinflammatory cytokines. (**A**) Timing of heat exposure and LPS/ATP stimulation in iBMMs. In each model, macrophages were treated with LPS (1 μg/mL) for 4 h, then ATP (5 mM) was added for 30 min to induce the release of IL-1β. Supernatants and cell lysate were analyzed by ELISA for cytokines and assayed for LDH release. One set of samples was incubated at 37 °C for 4 h as a control (black bars). Co-exposure of the “no priming” model: cells were stimulated with LPS and incubated at 41.5 °C or 39 °C for 1 h, followed by incubation at 37 °C for 3 h (hatched bars). “Priming” model: cells were stimulated with LPS and incubated at 37 °C for 3 h, followed by incubation at 41.5 °C or 39 °C for one hour (checked bars). (**B**–**D**) The co-exposure model at 41.5 °C shows inhibition of TNFα, IL-6, and intracellular IL-1β (hatched bars, 41.5 °C→37 °C). The primed model at 41.5 °C shows some inhibition of TNFα and IL-6 and complete inhibition of IL-1β (checked bars, 37 °C→41.5 °C). (**E**–**G**) 39 °C co-exposure model shows inhibition of IL-1β, TNFα, and IL-6 (hatched bars). The primed model at 39 °C shows inhibition of IL-1β, but not TNFα and IL-6 (checked bars). (**H**,**I**) The LDH-release assay shows cell death in the co-exposure model at both temperatures (hatched bars), but not in the primed model (checked bars). Results are the combined results of 2–3 independent experiments (IL-1β and LDH) or representative of 2–3 independent experiments (TNFα and IL-6). **** *p* < 0.0001 as determined by a two-way ANOVA with Tukey’s multicorrection comparison. *** *p* < 0.001, ** *p* < 0.01, ns—not significant.

**Figure 2 cells-12-01189-f002:**
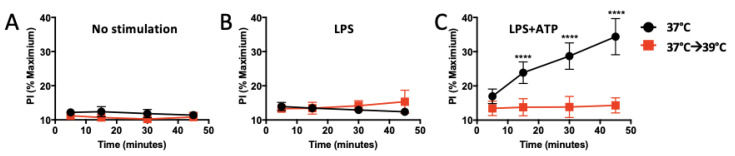
Mild heat shock inhibits membrane permeability. (**A**) unstimulated controls show no increased permeability: iBMMs were incubated at 37 °C (black circles) or exposed to 39 °C for one hour (red squares) followed by addition of PI (5 μM) to assay membrane permeability over time. (**B**) LPS controls show no increased permeability: iBMMs were primed with LPS and incubated at 37 °C for the entire 4 hours (black circles) or switched to 39 °C for the last hour (red squares), followed by addition of PI. (**C**) LPS+ATP stimulation shows that exposure to 39 °C inhibits ATP-induced pore formation in the membrane in LPS-primed cells: iBMMs were primed with LPS and incubated at 37 °C for the entire 4 hours (black circles) or switched to 39 °C for the last hour (red squares), followed by addition of ATP + PI The results are representative of two independent experiments. **** *p* < 0.0001 as determined by a two-way ANOVA with Tukey’s multicorrection comparison.

**Figure 3 cells-12-01189-f003:**
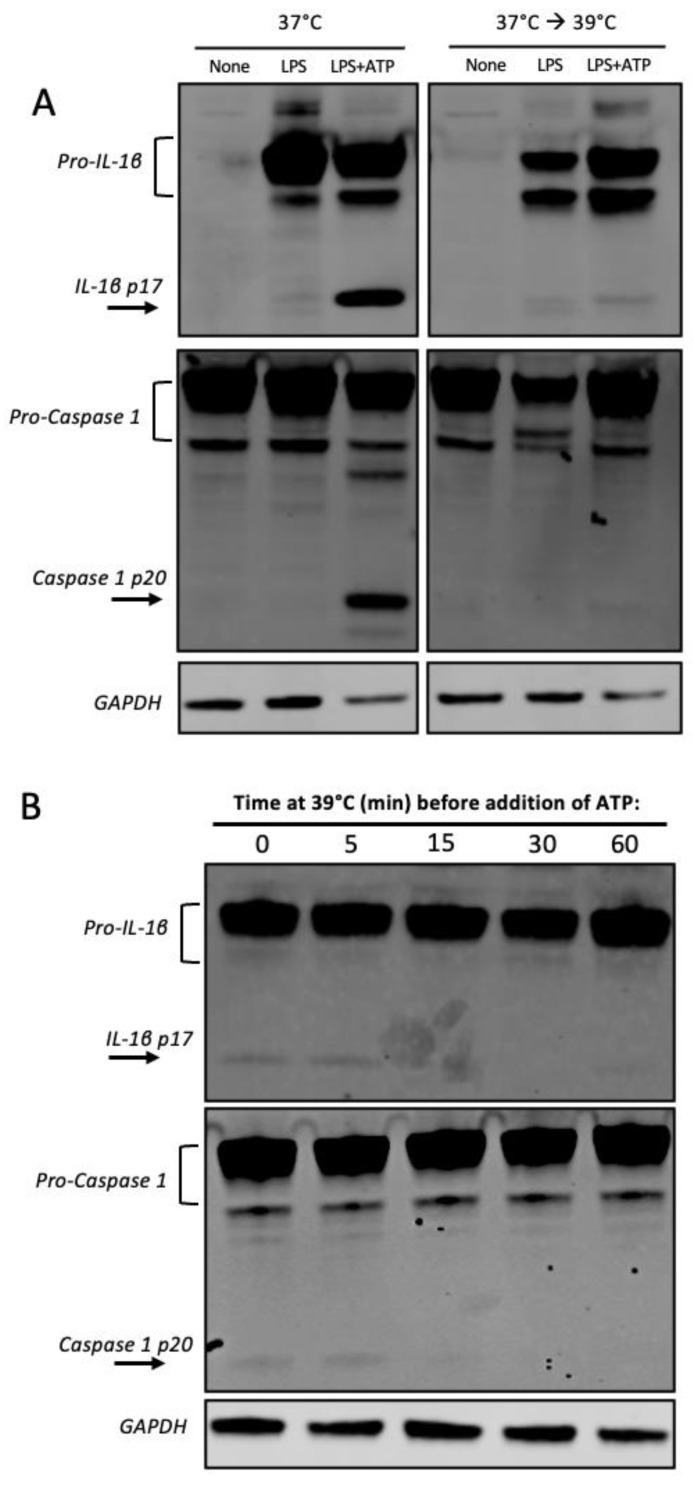
Mild heat shock rapidly inhibits caspase 1 and IL-1β processing. (**A**) Western blot shows presence of IL-1β processed fragment p17 and caspase 1 processed fragment p20 in LPS primed iBMMs treated with ATP incubated at 37 °C (Left panels), and decreased processing in heat-exposed cells (Right panels). (**B**) Time course: cells were primed for 3 h with LPS at 37 °C, then exposed to 39 °C for the indicated times, followed by addition of ATP to activate the processing of IL-1β. One of two independent experiments.

**Figure 4 cells-12-01189-f004:**
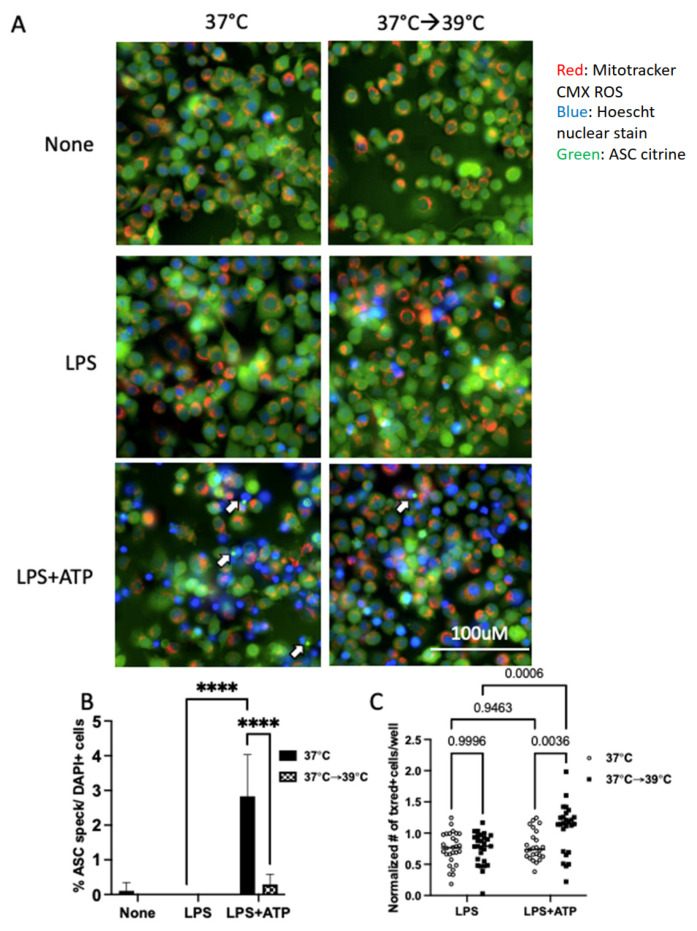
Mild heat shock inhibits the formation of ASC specks but does not inhibit mitochondrial membrane potential. (**A**) Representative images of iBMMs expressing ASC-citrine stained with Mitotracker CMX ROS and Hoescht nuclear stain show inhibition of ASC specks (white arrows) and increased numbers of cells with polarized mitochondria in heat-exposed cells. Live cells were imaged using the IXM live cell microscopy system. Images are from a representative experiment. (**B**) Quantification of ASC specks per DAPI nuclei. Results are a representative experiment of 3 independent experiments, counted manually with >5 (40×) fields/conditions. (**C**) Quantification of the number of cells/well with respiring mitochondria, normalized to the unstimulated 37 °C control condition, greater than 16 wells/condition/experiment combined over three independent experiments. **** *p* < 0.0001 as determined by a two-way ANOVA with Tukey’s multicorrection comparison; ns, nonsignificant.

**Figure 5 cells-12-01189-f005:**
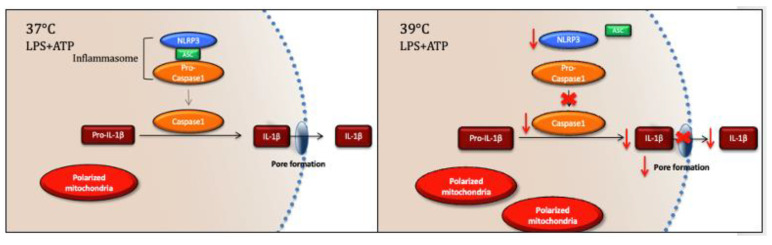
Exposure to a mildly elevated temperature of 39 °C leads to rapid inhibition of inflammasome activation, associated with decreased caspase 1 and IL-1β processing, decreased plasma membrane permeability, decreased ASC speck formation, and increased numbers of polarized mitochondria.

## Data Availability

The data presented in this study are contained within the article.

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
