# Peer review of "A Preliminary Study of Mild Heat Stress on Inflammasome Activation in Murine Macrophages"

_cells, 2023, doi:10.3390/cells12081189_

Round 1
Reviewer 1 Report
In this study, the authors showed that exposure to mild heat stress (39C for 15 24 minutes) rapidly inhibited iBMM inflammasome activity. Also, heat stress affect the mitochondrial respiration. The authors concluded that heat shock inhibits the NLRP3-dependent inflammasome.
Major comments
1) Figure 1: what about IL-18? Also, what the level of these cytokines at 37C.
2) Figure 3: Western blot for the followings are also required to confirm the findings : pro-caspase 1, prol-IL-1B, and NLRP-3
3) Figure 4: There is no significant difference between 37 and 39C. Why? also did the authors tried higher temp 41?
Author Response
Please see the attachment.
Reviewer 1: In this study, the authors showed that exposure to mild heat stress (39C for 15 24 minutes) rapidly inhibited iBMM inflammasome activity. Also, heat stress affects the mitochondrial respiration. The authors concluded that heat shock inhibits the NLRP3-dependent inflammasome.
We thank the Reviewer for the insightful comments and have addressed them as follows:
1. Figure 1, what about IL-18?
Since this was a preliminary study with limited scope, we chose to focus on IL-1b as the canonical proinflammatory cytokine and well-characterized readout for inflammasome activity. However, it would be a very good idea to look at IL-18, as it is another prominent cytokine in the IL-1 family dependent on caspase 1 processing. We will plan to examine this in a follow-up manuscript, and indicate this in our discussion.
2. Also, what the level of these cytokines at 37C.
The 37°C controls are now labeled with black bars. We have now adjusted the figure titles and legend as well as the text to clarify the experimental design and to indicate unambiguously the controls.
3. Figure 3: Western blot for the followings is also required to confirm the findings: pro-caspase 1, prol-IL-1B, and NLRP-3
We thank the reviewer for this suggestion and have added the western blots for pro-caspase 1 and pro-IL-1b. We have taken out the emphasis on heat exposure being a specific inhibitor of the NLRP3 inflammasome in particular, as we did not test for NLRP3 specificity in this study. However, we do now provide expression data for NLRP3 (Supplemental Figure 4S), showing that it is induced in our system by LPS+ATP and that heat exposure, even after LPS-priming, inhibits its expression.
4. Figure 4: There is no significant difference between 37 and 39C. Why? also did the authors tried higher temp 41?
We appreciate the reviewer’s prompt to re-look at this data. This data was generated from 3 high-throughput (96 well plate) experiments which varied in cell counts, so when presented in combination the data lost significance.
We reanalyzed this data by normalizing for cell counts in the unstimulated control condition for each experiment prior to combining. In doing so, it now shows a significant increase in numbers of polarized mitochondria.
In addition, we have added data for counts of ASC specs, finding that heat exposure inhibits formation of ASC specs.
5. also did the authors tried higher temp 41?
The overall scope of this paper is around the mild, potentially clinically relevant temperature of 39C; however, we have added this possibility to our future directions.

Reviewer 2 Report
This finding is very interesting. But i think authors this paper need additional data to demonstrate its finding. Since authors think NLRP3-dependent inflammasome plays a vital role, therefore, co-straining of NLRP3 and ASC proteins within one cell is required. In addition, i also suggested author could use caspase-1 inhibitor or NLRP3 knockdown to identify the function of NLRP3 inflammasome. This is very important.
Author Response
Reviewer 2: This finding is very interesting. But i think authors this paper needs additional data to demonstrate its finding. Since authors think NLRP3-dependent inflammasome plays a vital role, therefore, co-straining of NLRP3 and ASC proteins within one cell is required. In addition, i also suggested author could use caspase-1 inhibitor or NLRP3 knockdown to identify the function of NLRP3 inflammasome. This is very important.
We thank Reviewer 2 for the insightful comments and have addressed them as follows:
2.1) Since authors think NLRP3-dependent inflammasome plays a vital role, therefore, co-straining of NLRP3 and ASC proteins within one cell is required.
We agree that it would be important to indicate how we know heat inhibits the NLRP3 dependent inflammasome and not some other type of inflammasome (AIM, etc). Therefore, we have taken out the emphasis on NLRP3 from the title.
We have also added expression data for NLRP3 (Supplemental Figure 4S), showing that it is induced in our system by LPS+ATP and that heat exposure inhibits its expression.
In addition, we have now added data showing staining of ASC specs, with heat-treated cells showing reduced spec formation. In our future study it would be interesting to compare the heat effect on different types of inflammasome and confirm by staining and we have acknowledged this as a future focus in the text.
2.2) I also suggested author could use caspase-1 inhibitor or NLRP3 knockdown to identify the function of NLRP3 inflammasome.
We agree that for a mechanistic study it would be important to inhibit the phenotype once a mechanism is identified. As this was a preliminary observation, we did not fully address the mechanisms of how heat inhibits caspase 1 processing.
In our system it did not appear to be HSP72 (Supplemental Figure 5S), although a previous study did see this. One possibility is a distinct small HSP induced by heat is recruited to inhibit formation of the inflammasome, perhaps by NLRP3.
We have adjusted our text to clearly reflect these limitations. In our future mechanistic study, we would endeavor to explore whether such an HSP is recruited using pull-downs, use knock down (KD) or HSP inhibitors to confirm a role, and then, as you suggest, examine the effect of various NLRP3 KD /caspase-1 inhibitors on recruitment of this inhibitory HSP to the inflammasome complex.
In this preliminary study since we already see inhibition of caspase 1 processing with heat, using a caspase-1 inhibitor or NLRP3 KD would enhance the inhibition but we felt it might not add significantly to mechanism at this point.

Reviewer 3 Report
The authors Foster et al. have reported that the NLRP3-inflamatosome in macrophages is affected by heat shock stimulus, and macrophage activation is reduced following heat shock, resulting in the various cytokines development and releasing suppressed. The current results are exciting, and I have any concerns about publishing this article in this journal. But I have just one suggestion: a schematic illustration of the current results is beneficial to understand the authors' hypothesis. Therefore, please provide the summary diagram of the present results in Fig. 5.
Author Response
Reviewer 3: The authors Foster et al. have reported that the NLRP3-inflamatosome in macrophages is affected by heat shock stimulus, and macrophage activation is reduced following heat shock, resulting in the various cytokines development and releasing suppressed. The current results are exciting, and I have any concerns about publishing this article in this journal. But I have just one suggestion: a schematic illustration of the current results is beneficial to understand the authors’ hypothesis. Therefore, please provide the summary diagram of the present results in Fig. 5.
We thank the Reviewer for the thoughtful suggestion and have added a graphical abstract/Figure 5.
Reviewer 4 Report
Foster et al present a preliminary study on the effects of mild hyperthermia on cytokine production form an immortalised mouse macrophages cell line. While the hypothesis for the study is compelling and although the study is described as preliminary, I have several major concerns with the data. The experiments are not clearly presented and are missing basic controls. Overall, the study appears to have been very rushed and cannot be published in its current form. I outline my specific concerns with the data in each figure below:
Figure 1 is confusing with regard to IL-1b measurements – are these for intracellular or extracellular IL-1b? The text mentions intracellular but this is not clear from the figure. Inflammasome activation is characterised by the release of active IL-1b from the cell through GSDMD pores. The authors must measure extracellular IL-1b levels in order to determine inflammasome activity. In addition, NLRP3 inflammasome activation triggers caspase-1 activation and the cleavage of GSDMD which causes membrane pore formation and lytic cell death or pyroptosis. In Figure 1F no cell death is induced by LPS and ATP in the control temperature (37 degrees) this indicates that the inflammasome has not been activated and thus these results are not reliable.
Figure 2 PI is a membrane impermeant dye that is generally excluded from viable cells. It binds to double stranded DNA by intercalating between base pairs. It’s uptake thus indicates that the membrane has become permeabilised but does not indicate that GSDMD pores have been formed. PI uptake can also indicate apoptotic cell death. Pyroptosis is measured by LDH release assays (as in Figure 1) and GSDMD cleavage. The authors must show that pyroptosis is occurring tin their assays using these two measures.
Figure 3 These Western blot data are missing several key controls. Full length caspase-1 and pro-IL-1b must be shown and as per my previous comment GSDMD cleavage must also be measured. Given the focus on NLRP3, NLRP3 expression should also be measured. The full uncropped Western blots need to be included as supplemental data.
Figure 4 The stains used need to be indicated on the images. What is the green stain and what is the red stain? The methods describe ASC citrine but this is not mentioned in the text at all in reference to Figure 4. If ASC citrine has been used then some ASC specks can be visualised (indeed there appear to be some). These specks should be counted and quantified. The source of the ASC-citrine cells needs to be described in the methods.
Minor comments
Line 105 expressing should be removed or elaborated upon.
Line 135 REF Kagan add a reference or remove
Author Response
Reviewer 4: Foster et al present a preliminary study on the effects of mild hyperthermia on cytokine production form an immortalised mouse macrophages cell line. While the hypothesis for the study is compelling and although the study is described as preliminary, I have several major concerns with the data. The experiments are not clearly presented and are missing basic controls. Overall, the study appears to have been very rushed and cannot be published in its current form. I outline my specific concerns with the data in each figure below:
We thank the reviewer for the thorough and critical read of our manuscript and for providing insightful comments that we have addressed as follows:
4.1) Figure 1 is confusing with regard to IL-1b measurements – are these for intracellular or extracellular IL-1b? The text mentions intracellular but this is not clear from the figure. Inflammasome activation is characterised by the release of active IL-1b from the cell through GSDMD pores. The authors must measure extracellular IL-1b levels in order to determine inflammasome activity. In addition, NLRP3 inflammasome activation triggers caspase-1 activation and the cleavage of GSDMD which causes membrane pore formation and lytic cell death or pyroptosis. In Figure 1F no cell death is induced by LPS and ATP in the control temperature (37 degrees) this indicates that the inflammasome has not been activated and thus these results are not reliable.
We agree and have now re-labelled this figure to make it clear that both intracellular IL-1b and extracellular IL-1b are measured.
Following Evavold et al, 2018—intracellular (cell-associated) IL-1b was measured in the LPS only condition, and extracellular IL-1b measured in the supernatants after addition of ATP to activate the inflammasome. We are in alignment with reviewer 4 in finding that it is somewhat surprising that we were not able to measure significant lytic cell death by LDH release in the 37°C LPS+ATP control condition—this finding was consistent across more than 4 experiments.
However, given the release of IL-1b into the supernatant, the cleavage of pro-caspase 1 and pro-IL-1b (Figure 3), permeabilization of the membrane (Figure 2), and the formation of ASC specs (Figure 4), we are confident the inflammasome has been activated in the LPS+ATP control condition.
We are also now in agreement that to properly claim confirmed pyroptosis, that it would indeed be important to look at GSDMD, and as we could not accomplish this within the scope of this preliminary study, we have significantly adjusted the language and defocused from conclusions around pyroptosis, aspiring to obtain conclusive data on its presence in our future study.
4.2) Figure 2 PI is a membrane impermeant dye that is generally excluded from viable cells. It binds to double stranded DNA by intercalating between base pairs. It’s uptake thus indicates that the membrane has become permeabilised but does not indicate that GSDMD pores have been formed. PI uptake can also indicate apoptotic cell death. Pyroptosis is measured by LDH release assays (as in Figure 1) and GSDMD cleavage. The authors must show that pyroptosis is occurring tin their assays using these two measures.
Reviewer 4 are completely correct that to confirm pyroptosis we would need to see increased LDH in the control as well as increased PI uptake.
The increased PI uptake we observe in Figure 2 implies membrane permeabilization, but was not associated with necrotic cell death (as Figure 4 shows viable mitochondria). Therefore, it is likely we are activating the inflammasome at a level that is sufficient to cause pore formation but perhaps not pyroptosis (similar to hyperactivation).
Since we did not confirm GSDMD cleavage, as above, we have removed “pyroptosis” from our interpretation and adjusted the language throughout to better reflect the implications of our data.
4.3) Figure 3 These Western blot data are missing several key controls. Full length caspase-1 and pro-IL-1b must be shown and as per my previous comment GSDMD cleavage must also be measured. Given the focus on NLRP3, NLRP3 expression should also be measured. The full uncropped Western blots need to be included as supplemental data.
Thank you for asking for this data! We modified this figure to show full length caspase 1 and proIL-1b.
Full uncropped blots are available as supplemental.
Also, note additional, heat-dependent forms of caspase 1 become visible at approximately 40kD (Supplemental Figure 2S). These forms could potentially indicate heat induced altered processing, and we now, prompted by your thorough commentary, plan to further investigate this further in the future.
We have also added expression data for NLRP3 (Supplemental Figure 4S), showing that it is induced in our system by LPS+ATP and heat exposure inhibits its expression. We agree that it would be important to indicate how we know heat inhibits the NLRP3 dependent inflammasome – and thanks to your commentary have scaled back many claims resulting in what we feel is a more robust and useful paper.
4.4) Figure 4 The stains used need to be indicated on the images. What is the green stain and what is the red stain? The methods describe ASC citrine but this is not mentioned in the text at all in reference to Figure 4. If ASC citrine has been used then some ASC specks can be visualized (indeed there appear to be some). These specks should be counted and quantified. The source of the ASC-citrine cells needs to be described in the methods.
We have more clearly labeled this figure and added in the ASC quantification as requested, finding that heat shock inhibits ASC spec formation. We have also clarified the source of the ASC-citrine cells, which were immortalized in the Kagan lab from primary bone marrow macrophages derived from a transgenic ASC-citrine mouse from the laboratory of Douglas Golenbock (Evavold et al, 2018; Tzeng et al, 2016).
--
Minor comments
4.5) Line 105 expressing should be removed or elaborated upon.
This has been rectified accordingly
4.6) Line 135 REF Kagan add a reference or remove
This has been rectified accordingly
References.
Evavold, C.L.; Ruan, J.; Tan, Y.; Xia, S.; Wu, H.; Kagan, J.C. The Pore-Forming Protein Gasdermin D Regulates Interleukin-1 Secretion from Living Macrophages. Immunity 2018, 48, 35-44 e36, doi:10.1016/j.immuni.2017.11.013.
Tzeng, T.C.; Schattgen, S.; Monks, B.; Wang, D.; Cerny, A.; Latz, E.; Fitzgerald, K.; Golenbock, D.T. A Fluorescent Reporter Mouse for Inflammasome Assembly Demonstrates an Important Role for Cell-Bound and Free ASC Specks during In Vivo Infection. Cell Rep 2016, 16, 571-582, doi:10.1016/j.celrep.2016.06.011.

Round 2
Reviewer 1 Report
no further comment
Author Response
Thank you for the kind review of our manuscript!
Reviewer 2 Report
I think the revised version could be accepted.
Author Response

(The authors gave the same response as above.)

Reviewer 4 Report
The authors have rewritten the paper and added some experimental details which have somewhat improved the quality. However, I still have significant concerns about the data and cannot support publication of this study.
Major:
1. I have major concerns with the data presented in Figure 1. The lack of LDH release observed following LPS +ATP stimulation is highly inconsistent with the literature and my own experience working with these cells. Indeed in reference 16 clearly demonstrates that the iBMDM the authors received from the Kagan lab release LDH in response to NLRP3 stimulation. This suggests that the experimental set up the authors have used is not well controlled and therefore the data are not reliable. In addition there appears to be extreme variation in the levels of IL-1b that were detected between experiments (compare panel B 40,000 pg and panel E 2000 pg). Data from multiple experiments should be pooled rather than cherry picked as representative to show in a figure. With regards to the intracellular vs extracellular IL-1b levels how is it possible that there is more IL-1b processed and released than there is in LPS primed cells? These data suggest that perhaps this approach for determining intracellular pro-IL-1b levels is not reliable. Overall this indicates that there were problems with how the cells were cultured or how assays were performed and these data cannot be used to support the conclusions drawn by the authors.
2. I thank the authors for including the uncropped blots in Figure 3. However, I the supplemental data it’s concerning that the molecular weight indicated for pro-IL-1b is too high. The band is running above 35 kDa when the molecular weight of pro-IL-1b is 31 kDa. Indeed this uncropped blot also shows an interesting effect in the 39 ->37 treatment in the middle lanes where there is no pro-IL-1b in LPS treatment but there is with LPS and ATP. Can the authors explain these discrepancies? Perhaps they can include the blots for all of the experiments and not just show a representative. It is also unclear how many times the experiments in Figure 3 were performed. Is this N=1?
3. There appear to be a very low number of ASC specks in Figure 4 relative to the number of cells which is strange. ASC specks are usually counted per number of total cells, so expressed relative to the number of DAPI nuclei rather than field of view. Why is this number so low? Have these cells actually been activated?
4. Reference 10 is a study on alcohol inducing stress proteins and therefore does not support the statement that “Previous studies modeling hyperthermia in a dish indicated that exposing macrophages 72 in vitro to a heat shock of 42-43C can inhibit transcription of inflammatory cytokines 73 [9,10].” This reference should be changed or removed. All the other references should be checked by the authors to ensure they are correct and in fact support the statement they are making.
Minor:
1. Change ‘ASC spec’ to ‘ASC speck’
2. Change Triton X to Triton X-100
3. Details of ELISA kits used should be added ie. product codes
4. Details of the automated algorithm used in live cell counting should be included
